# Biochemical characterization of a GDP-mannose transporter from *Chaetomium thermophilum*

Gowtham Thambra Rajan Premageetha[1,2,3], KanagaVijayan Dhanabalan[1,2], Sucharita Bose[2], Lavanyaa Manjunath[2], Deepthi Joseph[2], Aviv Paz[4], Samuel Grandfield[4], Vinod Nayak[2], Luis M. Bredeston[5], Jeff Abramson[4], Subramanian Ramaswamy[1,2]*

1 Biological Sciences, Purdue University, West Lafayette, Indiana, United States of America, 2 Institute for Stem Cell Science and Regenerative Medicine, Bengaluru, Karnataka, India, 3 Manipal Academy of Higher Education, Manipal, Karnataka, India, 4 Department of Physiology, David Geffen School of Medicine at UCLA, Los Angeles, CA, United States of America, 5 Departamento de Química Biológica-IQUIFIB, Facultad de Farmacia y Bioquímica, Universidad de Buenos Aires-CONICET, Ciudad Autónoma de Buenos Aires, Junín, Argentina

* subram68@purdue.edu

**Data Availability Statement:** All relevant data are within the paper and its Supporting Information files.

## Abstract

Nucleotide Sugar Transporters (NSTs) belong to the SLC35 family (human solute carrier) of membrane transport proteins and are crucial components of the glycosylation machinery. NSTs are localized in the ER and Golgi apparatus membranes, where they accumulate nucleotide sugars from the cytosol for subsequent polysaccharide biosynthesis. Loss of NST function impacts the glycosylation of cell surface molecules. Mutations in NSTs cause several developmental disorders, immune disorders, and increased susceptibility to infection. Atomic resolution structures of three NSTs have provided a blueprint for a detailed molecular interpretation of their biochemical properties. In this work, we have identified, cloned, and expressed 18 members of the SLC35 family from various eukaryotic organisms in *Saccharomyces cerevisiae*. Out of 18 clones, we determined Vrg4 from *Chaetomium thermophilum* (CtVrg4) is a GDP-mannose transporter with an enhanced melting point temperature ($T_m$) of 56.9°C, which increases with the addition of substrates, GMP and GDP-mannose. In addition, we report—for the first time—that the CtVrg4 shows an affinity to bind to phosphatidylinositol lipids.

## Introduction

Glycosylation is the process that adds glycans to lipids and proteins. Most of these glycosylation reactions occur in the lumen of the endoplasmic reticulum (ER) and Golgi compartments. The building blocks for glycan biosynthesis are nucleotide sugars (NS), which function as substrates for glycosyltransferases to append sugar residues onto glycoproteins or glycolipids. In general, NS are synthesized in the cytosol, except CMP-sialic acid [1], and transported across the ER/Golgi membrane by Nucleotide Sugar Transporters (NST). NSTs function as

**Funding:** RS thanks support from DBT-B-life grant, Grant/Award Number: BT/PR5081/INF/156/2012, DBT-Indo Swedish Grant, Grant/Award Number: BT/IN/SWEDEN/06/SR/2017-18, ESRF Access Program of RCB, Grant/Award Number: BT/INF/22/SP22660/2017. AP and JA were supported by grant 5R35GM135175-03 from the National Institute of General Medical Sciences. Scientific-Technological Cooperation Program MINCyT-Argentina and DST-India (Grant Award Number IN/14/09 to LMB and RS). RS thanks support from SERB, India for Grant/Award Number EMR/2016/001825. KV thanks support from SERB, India, for a National post-doctoral fellowship. The funders had no role in study design, data collection, analysis, publication decision, or manuscript preparation. The authors declare that they do not have any competing interests.

**Competing interests:** The authors have declared that no competing interests exist.

antiporters where they transport nucleotide sugars into the lumen of ER/Golgi in exchange for nucleoside mono/di-phosphate (NMP/NDP) back to the cytosol for regeneration [2].

NSTs belong to the solute carrier SLC35 family of membrane transporters. This family is subdivided into seven subfamilies (SLC35A–G) that are further delineated by the specificity of the sugars they transport [3]. Humans have nine sugars—glucose (Glc), galactose (Gal), N-acetyl glucose (GlcNAc), N-acetyl galactose (GalNAc), glucuronic acid (GlcA), xylose (Xyl), mannose (Man), and fucose (Fuc)—conjugated to either GDP or UDP nucleotides. CMP-sialic acid is the lone monosaccharide available as a nucleotide monophosphate [4]. Although GDP-mannose is a naturally occurring NS in humans, no NST that transports it is found. Hence, it provides a unique opportunity to target GDP-mannose transporters for fighting fungal infections in humans where mannose is the most abundant sugar of the fungal cell wall, which directly supports the integrity of the cellSince NSTs serve as the primary transporters of NS, their loss of function has several consequences for human health and disease, resulting in Congenital Disorders of Glycosylation (CDG). Two well-documented autosomal recessive disorders linked to NSTs are leukocyte adhesion deficiency syndrome II [5] and Schneckenbecken dysplasia [6, 7], which results from a loss of function in the GDP-fucose (SLC35A1) and UDP-sugar transporters (SLC35D1) respectively. Additionally, NSTs have been linked to developmental disorders in invertebrates [8, 9] and pathogenicity and survival of lower eukaryotes [10]. Thus, a detailed structure and functional analysis are required.

After more than four decades of research on NSTs, the atomic resolution structure of the GDP-mannose transporter from *Saccharomyces cerevisiae* was determined in 2017 [11, 12]. More recently, NST structures of the maize CMP-sialic acid transporter [13] and the mouse CMP-Sialic acid transporter [14] have been determined. Despite their functional and sequence disparity, these NST structures reveal some common salient features. The crystal structures reveal that NSTs comprise ten transmembrane (TM) alpha helices where TM 1–5 is related to TM 6–10 via a pseudo twofold axis. As seen with many transporters, the inverted repeats share structural homology with little to no sequence similarities. To date, all structures of NSTs reside in an outward facing conformation (i.e., opening to lumen) where both substrates (nucleotide sugars and the corresponding NMP) bind to NST in a similar manner.

Lipids play a crucial role in altering NST's function, stability, conformation dynamics, and oligomeric state [15], yet no lipid-binding site(s) have been structurally resolved. A key aspect of NST's function is its interactions with lipids. Vrg4 from *S. cerevisiae* prefers short-chain lipids for its function [11]. Despite the similar structural architecture of the NSTs, it remains unclear the role lipids play in augmenting NSTs function and how minor changes in amino acid sequences correspond to sugar specificity. In this manuscript, we characterize a GDP-mannose transporter from *Chaetomium thermophilum* (CtVrg4) and screened several lipids for specific protein-lipids interaction. We show CtVrg4 prefers phosphatidylinositol species such as phosphatidylinositol-(3)-phosphate (PI3P), phosphatidylinositol-(4)-phosphate (PI4P), and phosphatidylinositol-(5)-phosphate (PI5P).

## Results

NSTs have proven to be difficult to express, purify, and structurally resolve. To overcome these limitations, we adopted the 'funnel approach' initially proposed by Lewinson *et al.*, 2008 [16]. We screened 18 NSTs, from different organisms, in an effort to find the ones more amenable to crystallization and characterization. Of these 18 constructs, we selected Vrg4 from *C. thermophilum* based on its expression, detergent extraction, and stability (S1 Fig and S1 Table) as a crystallization target. During the course of our work, the structure of the GDP-mannose transporter from *S. cerevisiae* (ScVRG4) was determined, causing us to refocus our priorities toward

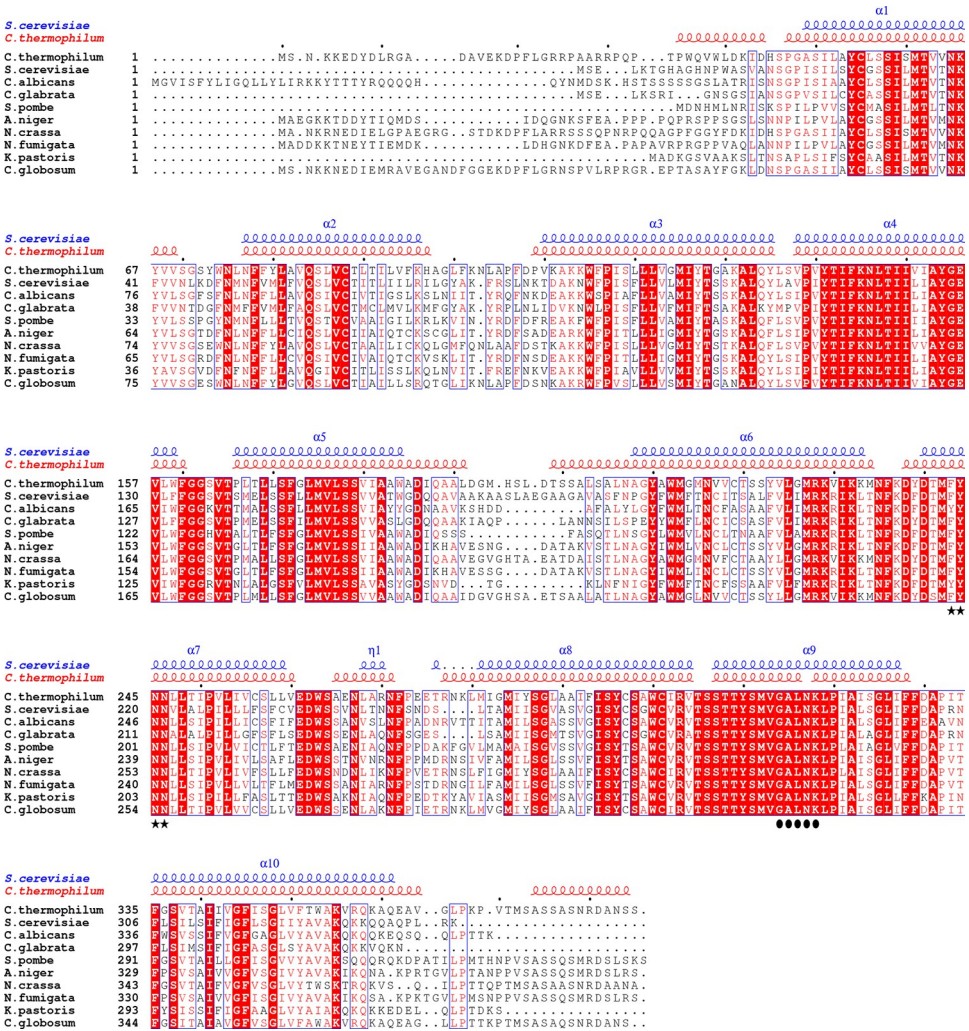

**Fig 1. Sequence alignment of *C. thermophilum* CtVrg4, *S. cerevisiae* ScVrg4 and other GDP-mannose transporter homologs.** Clustal Omega [17] was used for multiple sequence alignment of selected GDP-mannose transporters. Identical residues are highlighted in red, and highly conserved residues (>0.7) are highlighted in blue boxes. The nucleotide-binding and sugar recognition motifs are highlighted with star and closed circles, respectively, at the bottom of the alignment. The positions of the transmembrane domains are indicated and colored as blue and red, corresponding to ScVrg4 and CtVrg4 structures, respectively.

a detailed biochemical characterization of CtVrg4 [11]. The amino acid sequences of CtVrg4 and ScVrg4 are 53.6% identical (S2 Table), and both have the characteristic FYNN and GALNK GDP-mannose binding motifs (Fig 1). Due to these similarities, we generated a homology model for identifying structural components of CtVrg4 function.

## Chymotrypsin-cleaved CtVrg4 (cCtVrg4)

Initial crystallization trials were performed using a mosquito crystallization robot where purified CtVrg4 was equally mixed with 576 commercial crystallization screening conditions. Initial crystals were identified in 5 conditions and optimized by varying the contents of the crystallization components. Of these 5 conditions, the optimized condition of 0.07M sodium citrate pH 4.8, 75mM sodium fluoride, and 25% PEG300 yielded crystal that diffracted to 3.8Å. Unfortunately, these crystals proved difficult to reproduce and took almost a month for crystal

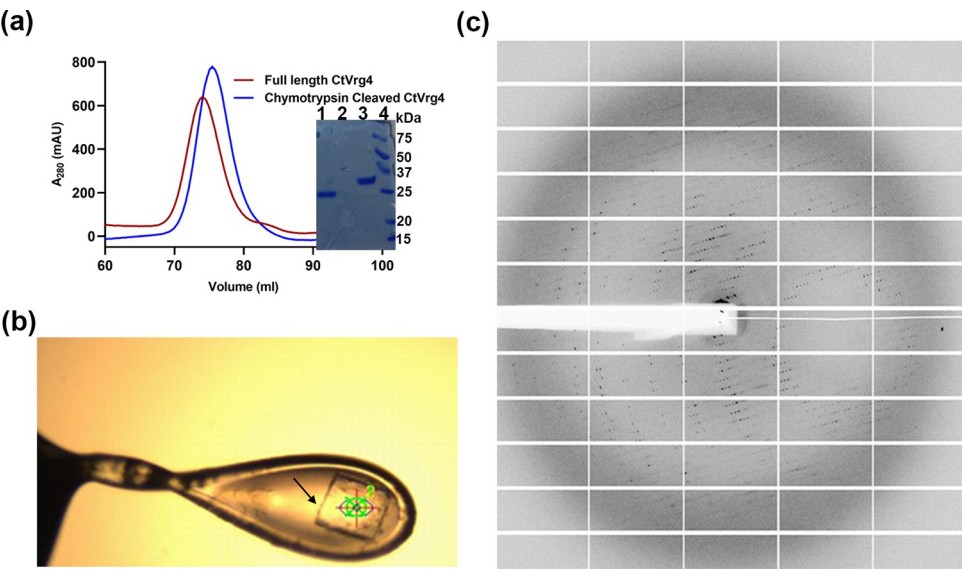

**Fig 2. Chymotrypsin cleaved CtVrg4 purification, crystallization, and diffraction pattern.** (**a**) Size exclusion profile of chymotrypsin cleaved CtVrg4 compared with full-length CtVrg4 protein. Inset shows SDS-PAGE of chymotrypsin cleaved CtVrg4 (lane 1) and full-length protein (lane 3), markers in lane 4. (**b**) CtVrg4 crystal indicated by the arrow. (**c**) Diffraction pattern of CtVrg4 crystal.

formation leading to difficulties in obtaining more detailed characterization of CtVrg4 crystals.

Further analysis of the CtVrg4 crystals by SDS-PAGE showed a significantly reduced molecular weight (~25 kDa) when compared to full-length purified protein, which has a molecular weight of 37 kDa (Fig 2A, gel insert) We speculated that CtVrg4 suffered from proteolysis and attempted to mimic this modification through incubation with chymotrypsin (cCtVrg4). After chymotrypsin induced proteolysis, cCtVrg4 elutes as a monodisperse peak at 75.44mL from the size exclusion column, where the full-length protein elutes at 74ml (Fig 2A). To test the functionality of cCtVrg4, we reconstituted the protein into liposomes made of Yeast Polar Lipid (YPL) and carried out transport assay. The proteoliposome transport assay showed that the cleaved protein is functional with a $K_m$ value of 32.07μM for GDP-mannose (S2 Fig). Additionally, cCtVrg4 crystalized reproducibly but did not diffract to high resolution for structure determination (Fig 2B and 2C).

## AlphaFold2 model of CtVrg4

We used AlphaFold2 to generate a model of the CtVrg4 [18] to aid in biochemical interpretation and to better assist structural comparisons with known NST structures [11, 13, 14]. The structural features are very similar and the confidence values of the prediction in the conserved regions are very high (S3 Fig).

The homology model shows the anticipated ten transmembrane helical structure corresponding to the NSTs fold. Additionally, the model predicted a 34 amino acids long stretch of disordered region and a short helix at the N-terminal and similarly a 17 amino acids long disordered region at the C-terminal of CtVrg4. This elongated stretch of disordered region is not seen in other NSTs (Fig 3A). Superimposition of the Alphafold2 model onto the ScVrg4 apo structure (PDB-5OGE Chain A) resulted in a good alignment of all the transmembrane helices with an RMSD value of 1.6Å (299 Cα atom pairs), barring unstructured and short helices at the terminus (indicated by arrowhead in Fig 3B).

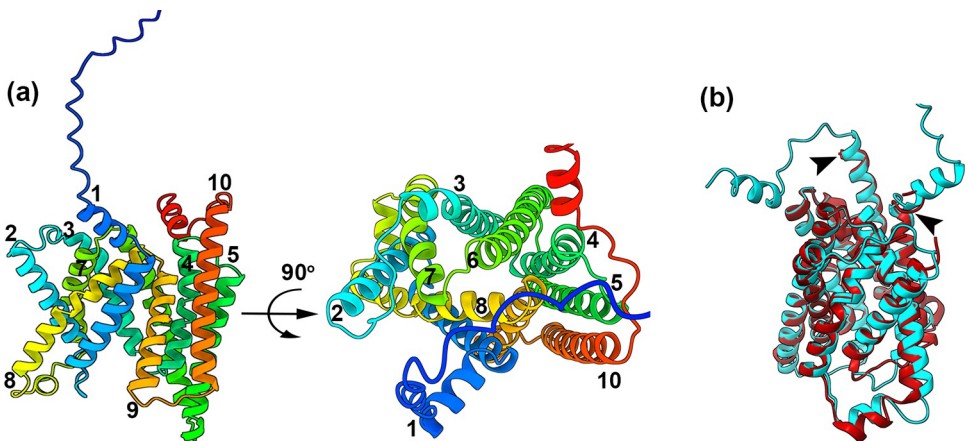

**Fig 3. AlphaFold2 model of CtVrg4. (a),** AlphaFold model of CtVrg4 with transmembrane helix colored from blue (N-terminal) to red (C-terminal) and numbered from 1 to 10. (**b**), Structural comparison of CtVrg4 model (Cyan) and ScVrg4 apo crystal structure (Maroon). Arrowhead indicates the unaligned structural element of the AlphaFold2 model with respect to the ScVrg4 structure.

## Complementation assay of CtVrg4

The functionality of CtVrg4 was assessed by complementation assay using a hygromycin B-sensitive yeast strain, NDY5, which lacks the Vrg4-2 gene [19]. Yeast that lacks or with an inhibited Vrg4 gene show defects in the glycosylation and the outer cell membrane becoming sensitive to hygromycin. This assay is a good proxy for the measurement of integrity of glycan structure. The assay shows that both CtVrg4 and ScVrg4 rescue NDY5 in hygromycin (Fig 4A). Additionally, based on the AlphaFold2 model CtVrg4Δ17(1−368 amino acids)—a truncation of 17 amino acids from the C-terminal end, which is a disordered loop region flanking the transmembrane helix 10 is able to rescue NDY5 in hygromycin. This indicates that the unstructured c-terminus is not necessary for function (Fig 4A).

Based on the ScVrg4 crystal structure (PDB-5OGK Chain A) published by Parker and Newstead [11], four residues—N220 and N221 form hydrogen bonds with the guanine moiety and Y28 and Y281 coordinate the ribose sugar (Fig 4B)—were identified for further functional characterization. In CtVrg4, alanine substitution of these amino acids shows either no (Y54A, Y310A) or only partial rescue (N245A, N246A) in the NDY5 assay (Fig 4A). More conservative substitution of Y310F and N246S also had limited ability for rescue. This complementation assay result agrees well with the *in vitro* transport assays of ScVrg4 mutants [11, 12] and establishes CtVrg4 as a GDP-mannose transporter.

## Transport kinetics of CtVrg4

Based on the ScVrg4 structure, the hydroxyl moiety of Y310 coordinates the ribose component of NS. To resolve the significance of the hydroxyl moiety in GDP-mannose binding and transport kinetics, we generated a protein with Y310F mutation. Proteins with CtVrg4 WT and Y310F mutation were reconstituted into the liposomes. The GDP-mannose $IC_{50}$ for WT CtVrg4 is 25.45μM and 47.05μM for Y310F (Fig 5A).

To further characterize CtVrg4 WT and Y310F mutant, a thermal shift assay was performed to determine the dissociation constant ($K_d$) in the presence of their substrates, GMP and GDP-mannose [20]. The $K_d$ values in Y310F mutant are similar for both substrates yielding a $K_d$ of 139.2μM for GMP and 129.95μM for GDP-mannose. However, the WT protein has a $K_d$

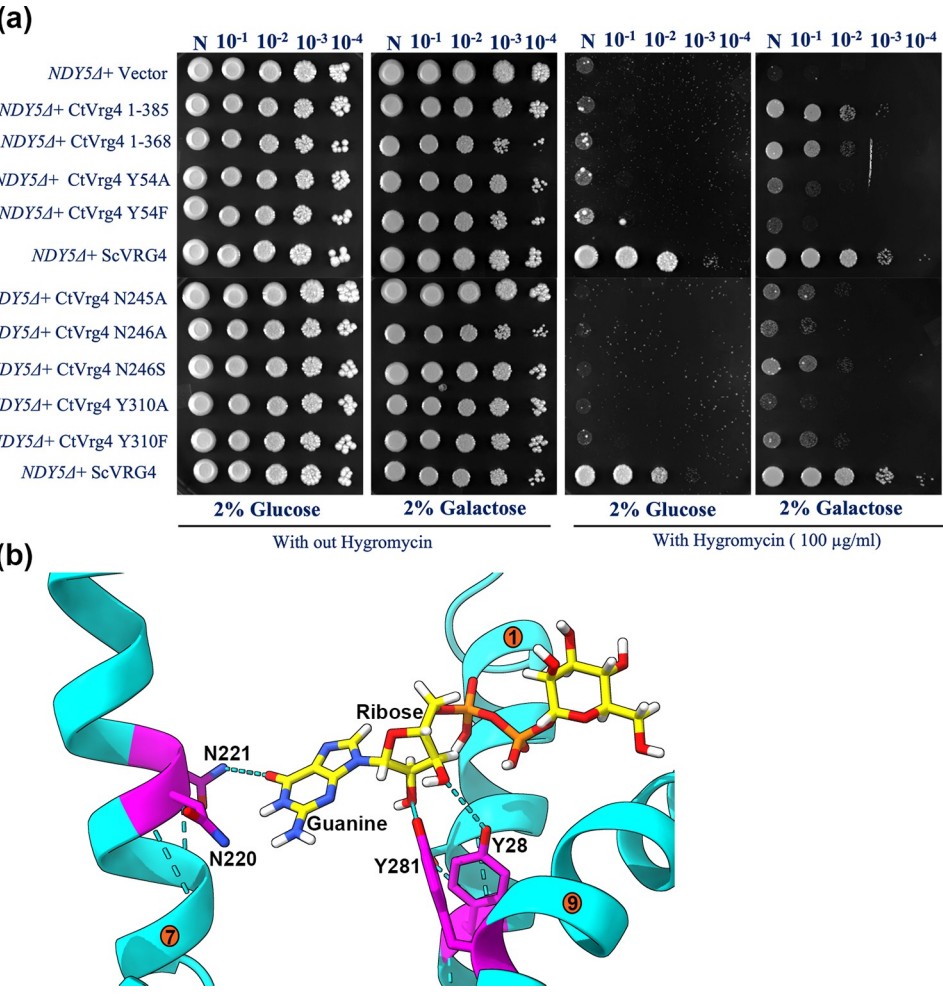

**Fig 4.** Functional Characterization of CtVrg4 and its mutants by a Hygromycin B based *in-vivo* assay: (**a**) Complementation assay of NDY5 by CtVrg4 WT and various functional mutants. Transformed yeast cells were serially diluted and spotted on synthetic agar media in the presence and absence of 100μg/mL of Hygromycin B. (**b**), Close-up view of ScVrg4 crystal structure. Equivalent amino acids that were chosen to be mutated in CtVgr4 are highlighted and shown in magenta stick color representation. GDP-Mannose is shown in yellow. Transmembrane helixes are numbered.

of 143.2μM for GMP but is reduced to 74.26μM for GDP-mannose (Fig 5B). Taken together, these suggests that the hydroxyl group is not critical for binding or transport of GDP-mannose by CtVrg4. We discuss later our idea that the importance of Y310 may come from its role in positioning Y54, which is critical for binding to the ribose sugar.

## Lipid binding activity and kinetics of CtVrg4 WT and CtVrg4Δ31 construct

Earlier studies showed that 1,2-dimyristoyl-sn-glycerol-3-phosphocholine (DMPC), is essential for ScVrg4 function [11]. We therefore screened several lipids for specific interaction with CtVrg4 wildtype protein using an established lipid blot assay [21]. In short, specific lipids are immobilized on strips (100 pmol) and subsequently bathed with purified protein to determine if there are protein/lipid interactions. After the incubation period, the strips are washed to remove nonspecific binding and the presence of protein is detected using antibody that recognizes the protein's poly-histidine affinity tag fused to the N-terminus. This assay revealed that

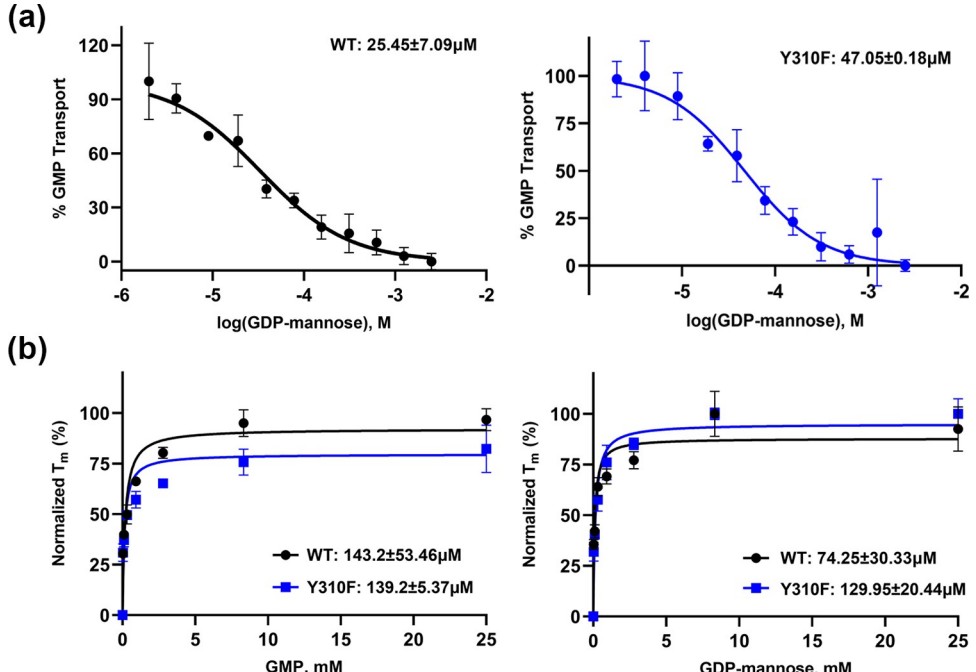

**Fig 5. Proteoliposome and thermal shift assay of CtVrg4. (a),** Representative GDP-mannose IC$_{50}$ curve for CtVrg4 WT and CtVrg4 Y310F. IC$_{50}$ values are shown at the top of the graph. **(b),** Thermal shift assay for CtVrg4 WT and CtVrg4 Y310F with varying concentrations of GMP and GDP-mannose. K$_d$ values are shown at the bottom of the graph. Calculated IC$_{50}$ and K$_d$ values are the means of two independent biological repeats (each done in technical duplicate or triplicate), errors are indicated as S.D.

CtVrg4 specifically binds to three phosphatidylinositol lipids—phosphatidylinositol-(3)-phosphate (PI3P), phosphatidylinositol-(4)-phosphate (PI4P), and phosphatidylinositol-(5)-phosphate (PI5P) (Fig 6A). These results suggest that CtVrg4 is possibly present in the Golgi membrane.

MD simulation by Parker *et al.* 2019 [12] predicted two lipid binding sites in the ScVrg4 protein. The first site is at the dimer interface formed by two transmembrane helices, TM5 and TM10, and the second is at the shallow groove between TM1, TM9, and TM10. Identifying TM10 as an integral component of lipid binding, we probed its' role by deleting the last 31 amino acid segments from the C-terminus, including the predicted Golgi retrieval signal (K$^{355}$VRQKA), which leaves most of the TM10 buried in the membrane (S3 Fig). This signal harbors several positively charged amino acids that could potentially bind to negatively charged phosphatidylinositol species. The new construct (CtVrg4Δ31) still showed binding for the same set of lipids as the WT protein, indicating the possibility of an additional lipid binding site.

This surprising result led to further characterization of the protein with CtVrg4Δ31 mutation. Both the proteoliposome and thermal shift assays for the truncated protein construct showed similar kinetics as the WT protein (Fig 6B and 6C). GDP-mannose IC$_{50}$ was found to be 48μM, and the K$_d$ for GMP and GDP-mannose was 63.2μM and 58.5μM, respectively. These results suggest that the C-terminal 31 residues are not critical for transport.

## Discussion

Our efforts to study the structure-function relationship of NST led to the identification of a thermostable GDP-mannose transporter, CtVrg4. CtVrg4 shares 53.6% sequence identity with

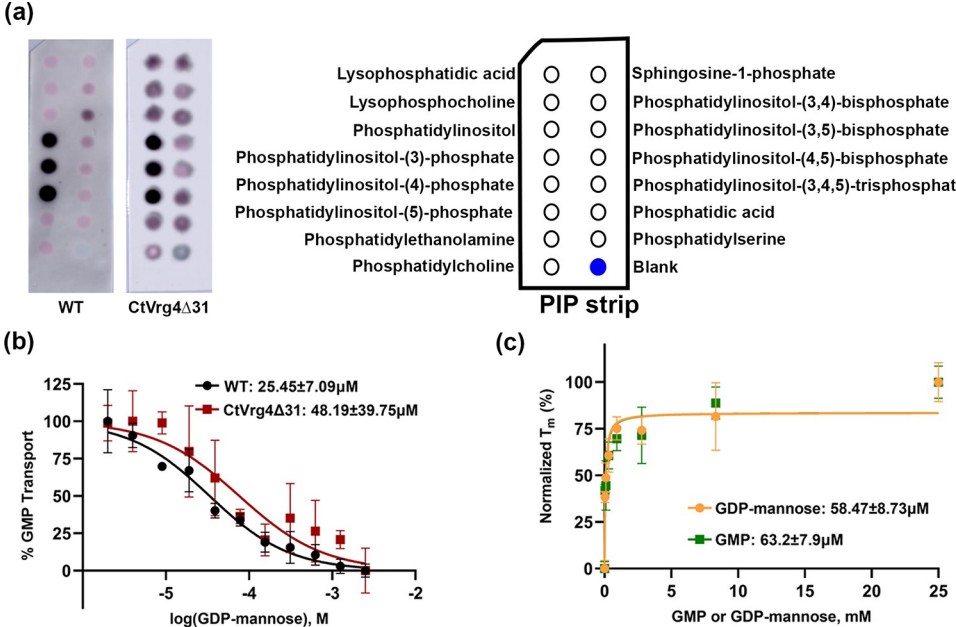

**Fig 6. Lipid binding activity and kinetics of CtVrg4Δ31. (a),** Lipid blot. Left–lipid binding activity of CtVrg4 WT and CtVrg4Δ31 protein. Right–legend with lipids present in the corresponding PIP strip spots. **(b),** Representative GDP-mannose $IC_{50}$ curve for CtVrg4Δ31. The $IC_{50}$ value is shown at the top of the graph. **(c),** Thermal shift assay for CtVrg4Δ31 with varying concentrations of GMP and GDP-mannose. $K_d$ values are shown at the bottom of the graph. Calculated $IC_{50}$ and $K_d$ values are mean of two independent biological repeats (each done in technical duplicate or triplicate), errors are indicated as S.D. Lipid binding assay was done in at least two biological replicates.

ScVrg4 and maintains the conserved substrate binding motifs, FYNN and GALNK, indicative that CtVrg4 is a GDP-mannose transporter as well. We further confirmed that CtVrg4 is a GDP-mannose transporter utilizing a hygromycin-based complementation assay and substantiated our finding by *in vitro* proteoliposome transport assays. The $IC_{50}$ value of GDP-mannose was found to be 25.45μM, which is approximately three times higher than that for ScVrg4 (7.7μM). Our melting point assay in DDM showed CtVgr4 is significantly more stable (56.9°C) (Fig 7A) than ScVrg4 (37.9°C). This is not surprising as CtVrg4 is isolated from a thermostable fungus.

During crystallization trials, we found that the chymotrypsin cleaved CtVrg4 produced better diffraction quality crystals than the full-length protein. Proteoliposome transport assay of chymotrypsin cleaved CtVrg4 showed that the cleaved protein is functional with a GDP-mannose $K_m$ value of 32.07 μM, suggesting that limited proteolysis does not affect the core of the protein. The observation that a shorter construct is still functional suggests it might be worth reattempting the structure determination, which we did not pursue further after the structure of ScVrg4 was published. Molecular replacement with the 3.8 Å data did not produce good solutions. Given the similarity in sequence between CtVrg4 and ScVrg4 (53%), and the modern structure prediction tools like AlphaFold2, the predicted structure would be a good model (compared to a 3.8 Å structure).

In ScVrg4, Y281 forms a hydrogen bond with the ribose sugar of GDP-mannose and a π-stacking interaction with Y28, which along with S269, further coordinates the ribose sugar. The alanine mutants Y28A, and Y281A of ScVrg4, resulted in the complete abolishment of transport activity [11]. In our complementation assay, we found that Y310F (equivalent to Y281 in ScVrg4) conferred a partial rescue, whereas no complementation was observed in the

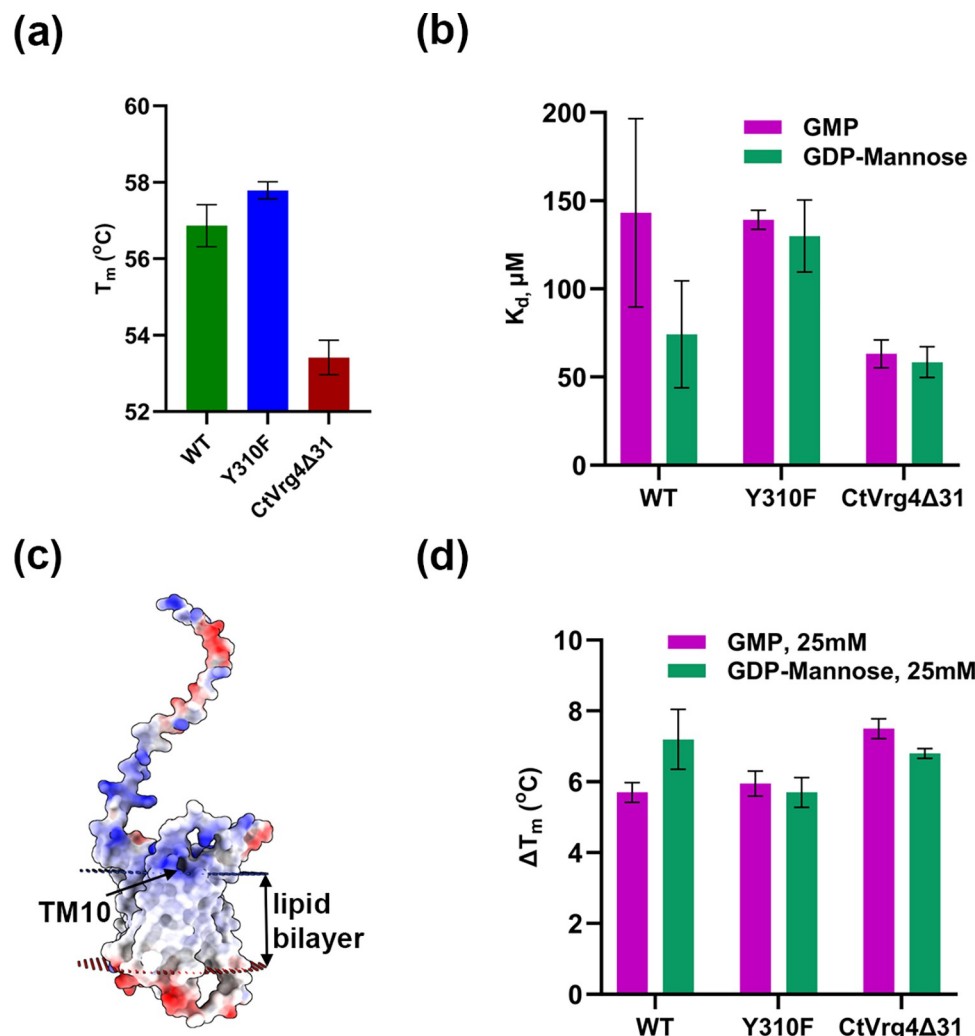

**Fig 7. Thermal stability analysis of CtVrg4. (a),** Bar graph of melting temperature ($T_m$) of CtVrg4 WT, Y310F, and CtVrg4Δ31. **(b),** $K_d$ values of GMP/GDP-mannose binding to CtVrg4 WT, Y310F, and CtVrg4Δ31 were estimated by a thermal shift assay. **(c),** Electrostatic surface representation of the AlphaFold2 model for CtVrg4 highlighting positively charged residues in the Golgi retrieval sequence in TM10. The orientation of CtVrg4 in the lipid bilayer was simulated through the OPM server [22] **(d),** GMP/GDP-mannose binding to CtVrg4 was estimated by a shift in $T_m$ after the addition of 25 mM GMP/GDP-mannose to WT, Y310F, and CtVrg4Δ31.

case of Y310A, Y54A, and Y54F. The $K_d$ for GDP-mannose almost doubled in the Y310F mutant compared with WT, but the $K_d$ for GMP remained the same (Fig 7B). The $IC_{50}$ for GDP-mannose also doubled compared with the WT, further validating the decreased binding efficiency of GDP-mannose in the Y310F mutation. These results suggest that Y310 also forms a hydrogen bond with the ribose sugar and helps position the Y54 residue to bind the ribose sugar, whose substitution is lethal for transport activity.

It was previously reported that ScVrg4 localizes to the Golgi using its C-terminal Golgi retrieval sequence, which binds to COPI vesicles [23]. CtVrg4 has a Golgi retrieval sequence ($K^{355}$VRQKA) like ScVrg4 in the cytoplasmic end of TM10. The Golgi retrieval sequence is positioned on membrane boundaries to interact with the membrane lipids (Fig 7C). The Golgi retrieval sequence is rich in positively charged residues, which could be a potential binding site for negatively charged PIP lipids to bind. However, our results show that CtVrg4Δ31 (devoid

of the last 31 amino acids from the C-terminal end, including the Golgi retrieval sequence) has the same binding affinity as the full-length protein—suggesting the Golgi retrieval sequence may not play any role in PIP binding. Further studies are needed to pinpoint the exact location of PIP binding to the transporter and its structural or functional effect on CtVrg4.

We report that CtVrg4Δ31 is less thermostable than the WT and Y310F mutant with a 3–4˚C lower melting temperature (Fig 7A). Interestingly, both GMP and GDP-mannose bind to CtVrg4Δ31 and show similar thermal shift values compared with WT and Y310F (Fig 7D). This result suggests that although overall stability is affected by truncation, the ability of the substrate to bind and stabilize the core protein is not affected. Our work reported here suggests that further structure function studies of NSTs are needed in order to understand the mechanism of transport and the role of lipids in localization, stability and NS transport.

## Materials and methods

### Cloning and expression of CtVrg4 wildtype, mutants, and truncated constructs in a yeast expression system

The sequence of CtVrg4 from *C. thermophilum* (NCBI GenBank XM_006692792.1) was identified through a homology search against the ScVrg4 sequence. CtVrg4 is 385 amino acids long. The gene was cloned and expressed in a modified pDDGFP yeast expression vector with an N-terminal 12 histidine tag followed by a SmaI restriction site for homologous recombination-based gene insertion and a stop codon.

All CtVrg4 mutants were generated via site-directed mutagenesis using the PCR method and further confirmed by sequencing. Two C-terminal truncated versions of CtVrg4 were constructed, CtVrg4Δ17 (constituting amino acids from 1 to 368) and CtVrg4Δ31 (comprising amino acids from 1 to 354). The cloned genes were expressed in *S. cerevisiae* haploid strain FGY217 (*MATa*, *ura3-52*, *lys2_201*, and *pep4*).

### Expression and purification of CtVrg4

The primary cultures of CtVrg4 WT, mutations, and truncations were grown in synthetic media without uracil in 2% glucose. The primary culture was diluted into a secondary culture (1L) to the final OD of 0.2 in 0.1% glucose. The culture was induced with 2% galactose at 0.6 to 0.8 OD to express the protein and further grown for 22–24h before harvesting the cells by centrifugation. Cells were resuspended in membrane resuspension buffer (75mM Tris pH 8.0, 150 mM NaCl, and 5% glycerol) and then lysed using a cell disruptor (Constant Systems Ltd) at 28, 32, 36, and 39 kpsi. Membranes were isolated by centrifugation at 200,000 xg for 1.5h. The protocol for protein purification was adapted from Drew *et al*., 2008 with modifications [24]. The membranes were solubilized in dodecyl β-D-maltopyranoside (DDM, Anatrace) at a 1: 0.2 (w/w) ratio of the membrane to DDM for 2 h in membrane resuspension buffer along with a protease inhibitor cocktail. For crystallization, the protein was solubilized in 2% Decyl β-D-maltopyranoside (DM) for two hours in the membrane resuspension buffer. The solubilized membrane was centrifuged at 200,000 xg for 30 minutes, and the supernatant was loaded onto a 5 mL His-Trap FF/HP column (Cytiva). The protein was eluted with 300 mM imidazole. The eluted fractions were desalted using a Hiprep 26/10 desalting column (Cytiva) to remove the imidazole from the buffer. The protein was concentrated using a 50 kDa cut-off Amicon device (Millipore). The concentrated protein was centrifuged at 14,000 xg for 15 mins before injecting onto a Superdex 200 (10/300) size-exclusion column (Cytiva) pre-equilibrated with liposome assay buffer (20 mM HEPES pH 7.4, 50 mM KCl, and 2 mM $MgSO_4$) containing 0.1 mM EDTA and 0.013% DDM.

## Chymotrypsin cleavage and crystallization

Purified CtVrg4 was subjected to limited proteolysis using chymotrypsin (50:1 ratio) for 2h at 18°C. The reaction was stopped with 0.1mM phenylmethylsulfonyl fluoride followed by ultra-centrifugation at 150,000 xg for 30 minutes. The chymotrypsin cleaved protein eluted at 75.44ml compared to 74ml for the full-length protein on a Superdex 200 (16/600) column (Cytiva). The cleaved protein's peak fractions were pooled and concentrated using 50kDa cut-off Amicon (Millipore).

Purified chymotrypsin cleaved CtVrg4 (12mg/ml) was mixed in a 4:1 (protein/bicelle) ratio with a 25% (2.8:1) DTPC/CHAPSO bicellar solution for 45 minutes on ice, yielding 9.6 mg/ml of chymotrypsin cleaved CtVrg4 in 5% bicelles. The best crystals grew to a size of 0.07 mm X 0.07 mm X 0.02 mm at 18°C in 0.07 M sodium citrate pH 4.8, 75 mM sodium fluoride, and 25% PEG300. The crystals were cryoprotected with 30% PEG400 before flash-cooling in liquid nitrogen. X-ray diffraction data were collected at the PROXIMA-1 beamline, SOLEIL synchrotron source (France), at a wavelength of 0.97857 Å using a 100 K nitrogen stream. The best crystal diffracted to 3.8 Å resolution.

## Protein reconstitution into liposomes

Chymotrypsin cleaved CtVrg4 was reconstituted into yeast polar lipid (YPL, Avanti Polar Lipids) liposomes in 20 mM HEPES pH 7.5, 100 mM KCl, as described by Parker *et al.*, 2017 [11]. CtVrg4 wildtype/Y310F/truncates were reconstituted into liposomes using a modified protocol from Majumdar *et al.*, 2019 [25]. Briefly, YPL were suspended in chloroform, dried using a nitrogen stream, and left in a vacuum desiccator overnight. The lipid film was resuspended at 10 mg/mL in liposome assay buffer, and then bath sonicated to form small multilamellar vesicles. Vesicles were extruded for 10 cycles through 400 nm polycarbonate membranes (Avanti Polar Lipids). For reconstitution, purified (CtVrg4 wildtype/Y310F/truncate) protein in DDM (at 8 to 16 mg/mL) was added to the extruded YPL at a final lipid: protein ratio (w/w) of 80:1. Sodium cholate (Anatrace) was added at a concentration of 0.65–0.75% to this mixture and incubated for one hour at room temperature, then for a further 30 minutes on ice. As a control, liposomes without protein were resuspended in the assay buffer containing 0.013% of DDM. The protein-lipid mixture was passed through an 8.3 mL PD10 column (Cytiva) pre-equilibrated with 0.5 mg YPL in liposome assay buffer. A fraction volume of 2.8 mL was collected after the column's void volume (2.6 mL) and centrifuged at 150,000 xg for 30 minutes. Subsequently, the pellet containing proteoliposomes was resuspended with liposome assay buffer, flash-frozen in liquid $N_2$, and stored at -80°C. The protein reconstitution into the lipids was verified by Western blot with anti-His antibody.

## Transport assay

CtVrg4 wildtype/Y310F/truncate proteoliposomes were thawed, and the desired concentration of internal cold substrate (1 mM GDP-mannose) was added and then subjected to six rounds of freeze-thaw in liquid nitrogen to load the proteoliposomes with the substrate. For chymotrypsin-cleaved CtVrg4, the internal concentration of GDP-mannose ranged from 1 to 1000 µM. Unloaded GDP-mannose was removed by ultracentrifugation at 150,000 xg for 30 minutes. The pellets were resuspended in a cold liposome assay buffer. For a typical 50 µl reaction, 10 µl of loaded proteoliposomes containing approximately 4 µg of protein was added to 40 µl of liposome assay buffer containing 0.5 µM [3H]GMP (American Radiolabeled Chemicals, Inc). For the chymotrypsin cleaved protein in proteoliposomes, the concentration of [3H]GMP was 0.384 µM. The addition of GMP initiated the exchange reaction. For $IC_{50}$ value determination, the external competing substrate was varied from 2 µM to 2.5 mM. The

mixture was then incubated at 25˚C for 20 minutes. For chymotrypsin cleaved CtVrg4, the reaction was performed at room temperature and terminated after 10 min. The uptake of the radiolabeled substrate was stopped by adding 800 μl of cold water and rapidly filtering onto 0.22 micron mixed cellulose esters filters (Millipore), which were then washed three times with 2 mL of ice-cold water. A liquid scintillation counter (Perkin Elmer) measured the amount of [3H]GMP transported inside the liposomes. All experiments were done in two biological repeats, each in technical duplicate or triplicate. Kinetic parameters were calculated by nonlinear fit using the GraphPad Prism software (GraphPad Software, Inc., San Diego, CA, USA).

## Hygromycin B-based *in vivo* assay

For *in vivo* functional characterization, wildtype/mutant/truncated proteins were expressed in the yeast NDY5 strain (MAT *ura3–52a*, *leu2–211*, *vrg4-2*), which is a *vrg4Δ* mutant. The transformants were selected based on the uracil resistance marker. Cells grown overnight were serially diluted and spotted on the synthetic agar media in the presence and absence of 100 μg/mL of Hygromycin B. Protein expression was induced with 2% galactose, and phenotype was observed after three days. For negative control, transformed cells were spotted in 2% glucose and compared with 2% galactose.

## Lipid blot assay

Lipid blot assay was carried out using PIP Strips™ (Echelon Biosciences) following the manufacturer's instructions. Briefly, the strips were blocked overnight with PBST buffer (1x PBS with 0.05% Tween-20) containing 5% non-fat dry milk at 4˚C with mild rocking. The strips were then incubated with 50 μg of CtVrg4 WT/CtVrg4Δ31 in PBST for two hours at room temperature. After three washes with PBST, the strips were incubated with conjugated anti-polyHistidine–Peroxidase antibody (Sigma A7058, in a 1:2000 ratio) for one hour at room temperature. After three more washes with PBST, the bound proteins were detected with the Clarity™ Western ECL kit (Bio-Rad Laboratories).

## Thermal shift assay

Following size exclusion chromatography, detergent-solubilized CtVrg4 wildtype/mutant/truncate (final concentration of 0.5mg/mL) were incubated with 0 mM to 50 mM GMP/GDP-mannose at room temperature for 15 minutes before measuring the $T_m$ using a Tycho NT.6 instrument (NanoTemper Technologies, Germany). All experiments were done in two biological repeats and three technical repeats. Kinetic parameters were calculated by nonlinear fit using the GraphPad Prism software.

## AI prediction and superimposition

The AlphaFold2 structure prediction feature in UCSF ChimeraX was used to predict the structure of CtVrg4. Superimposition and all protein model figures were generated with UCSF ChimeraX [26]. ESPript was used to render sequence similarities and secondary structure information onto multiple sequence alignments [27].

## Supporting information

**S1 Fig. A schematic of the funnel approach that was used to screen for crystallizable NSTs.**
(TIF)

**S2 Fig. Exchange velocities of internal GDP-man with external [3H]GMP in proteoliposomes.** Values are the means of three independent biological repeats (each done in technical

triplicate). Errors are indicated as SD. $K_m$ was calculated by non-linear fit using the GraphPad Prism software.
(TIF)

**S3 Fig. AlphaFold2 model for CtVrg4.** Colors represent the predicted confidence value of the structure. The confidence value decreases as the color changes from red to blue. The figure shows that the transmembrane regions are predicted with high confidence–in red. The N-terminus region is floppy and is predicted poorly. Note that the C-terminal helix is predicted with intermediate confidence (green). Amino acid Alanine 27 (A27) and the C-terminal residue (S385) are labeled for reference. A354, is the last residue in the CtVrg4Δ31 construct.
(TIF)

**S1 Table. NSTs identified for crystallization through homology search.**
(DOCX)

**S2 Table. Percentage sequence identity of known and putative GDP-mannose transporters from different species of fungi—alignment carried out using Clustal Omega.**
(DOCX)

**S1 Raw image.**
(PDF)

# Acknowledgments

We thank Dr. Leonard Chavas from SOLEIL Synchrotron (Proxima-1 beamline) for providing the beamtime for data collection. The Tycho experiment was done in the Chemical Genomics Facility at Purdue Institute for Drug Discovery and the NIH-funded Indiana Clinical and Translational Sciences Institute.

# Author Contributions

**Conceptualization:** KanagaVijayan Dhanabalan, Luis M. Bredeston, Jeff Abramson, Subramanian Ramaswamy.

**Data curation:** Gowtham Thambra Rajan Premageetha, KanagaVijayan Dhanabalan, Jeff Abramson.

**Formal analysis:** Gowtham Thambra Rajan Premageetha, KanagaVijayan Dhanabalan, Sucharita Bose, Lavanyaa Manjunath, Vinod Nayak.

**Funding acquisition:** Luis M. Bredeston, Jeff Abramson, Subramanian Ramaswamy.

**Investigation:** Gowtham Thambra Rajan Premageetha, KanagaVijayan Dhanabalan, Sucharita Bose, Lavanyaa Manjunath, Deepthi Joseph, Aviv Paz, Samuel Grandfield, Vinod Nayak, Luis M. Bredeston, Jeff Abramson, Subramanian Ramaswamy.

**Methodology:** Gowtham Thambra Rajan Premageetha, KanagaVijayan Dhanabalan, Sucharita Bose, Lavanyaa Manjunath, Vinod Nayak.

**Supervision:** Luis M. Bredeston, Jeff Abramson, Subramanian Ramaswamy.

**Validation:** KanagaVijayan Dhanabalan.

**Writing – original draft:** Gowtham Thambra Rajan Premageetha, KanagaVijayan Dhanabalan, Jeff Abramson, Subramanian Ramaswamy.

**Writing – review & editing:** Gowtham Thambra Rajan Premageetha, KanagaVijayan
Dhanabalan, Sucharita Bose, Lavanyaa Manjunath, Deepthi Joseph, Aviv Paz, Vinod
Nayak, Luis M. Bredeston, Jeff Abramson, Subramanian Ramaswamy.

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
