## [Decision Letter · Decision Letter 0]

21 Feb 2023

PONE-D-23-00914Biochemical characterization of a GDP-mannose transporter from Chaetomium thermophilumPLOS ONE

Dear Dr. Subramanian,

        Thank you for submitting your manuscript to PLOS ONE. I apologize for the tardiness in the review process. After careful consideration, we feel that it has merit but does not fully meet PLOS ONE’s publication criteria as it currently stands. Therefore, we invite you to submit a revised version of the manuscript that addresses the points raised during the review process.

      Despite one reviewer noting minor changes and the other recommending major changes, you will see from the comments that there are major concerns by both reviewers that I agree with.

       Please submit your revised manuscript by Apr 07 2023 11:59PM. If you will need more time than this to complete your revisions, please reply to this message or contact the journal office at plosone@plos.org. Please include the following items when submitting your revised manuscript:A rebuttal letter that responds to each point raised by the academic editor and reviewer(s). You should upload this letter as a separate file labeled 'Response to Reviewers'.A marked-up copy of your manuscript that highlights changes made to the original version. You should upload this as a separate file labeled 'Revised Manuscript with Track Changes'.An unmarked version of your revised paper without tracked changes. You should upload this as a separate file labeled 'Manuscript'.

We look forward to receiving your revised manuscript.

Kind regards,

Michael Massiah

Academic Editor

PLOS ONE

Journal Requirements:

" ext-link-type="uri" xlink:type="simple">https://journals.plos.org/plosone/s/file?id=ba62/PLOSOne_formatting_sample_title_authors_affiliations.pdf"

Reviewers' comments:

Reviewer's Responses to Questions

**Comments to the Author**

1. Is the manuscript technically sound, and do the data support the conclusions?

Reviewer #1: Partly

Reviewer #2: Yes

2. Has the statistical analysis been performed appropriately and rigorously? 

Reviewer #1: No

Reviewer #2: Yes

3. Have the authors made all data underlying the findings in their manuscript fully available?

Reviewer #1: Yes

Reviewer #2: Yes

4. Is the manuscript presented in an intelligible fashion and written in standard English?

Reviewer #1: Yes

Reviewer #2: Yes

5. Review Comments to the Author

Reviewer #1: The work by Gowtham Thambra Rajan Premageetha et al. explores the function of CtVrg4 nucleotide sugar transporter, after an attemp to characterize its crystalized structure. The text describe in a honest way the different approaches used to study this NST after the publication of the crystalized structure of ScVrg4. The assays and results are described in a clear manner, although no clear rationale is observed across the different experiments, in order to understand why each assay was performed or how each results connects to another. For example, why not analyze the CtVrg4 cleaved form (cCtVrg4) on the complementation assay with Hygromycin B or the CtVrg4d31 version. In the same way, why do not probe the CtVrg4d17 on the lipid assay in parallel to the CtVrg4d31 deletion mutant. Independent of these comments, the major drawback of this work is the lack of biological replicates in the assays described in the figures 5, 6 and 7. Since these experiments lack of biological replicates, authors does not report statical analysis as expect but it impact on the overall interpretation of results. It would be adequate to include more biological replicates and perform an statical analysis to support these observations as the journal requires. All other aspect meet the journal requirements.

Reviewer #2: The authors characterized CtVrg4 as new GDP transporter from fungus and is well compared with known GDP transporter from Yeast ScVrg4. CtVrg4 established to be similar structure as ScVrg4, However the manuscript explains the differences in their kinetics and lipid binding of these two proteins. The experiments are studied by exploring its various mutants and shows that C-terminal part of protein is not essential for its function and binding to specific lipids.

I have following concerns regarding manuscript:

Major points

1. Line 170: Based on binding studies, the authors shows that hydroxyl group Y310 in CtVrg4 is not critical for binding. It makes sense for binding of GMP-mannose with CtVrg4 WT /mutant Y130F as Kd is in same range but for binding of GDP with CtVRg4 WT is stronger compared to mutant as Kd is much lower. The hydroxyl group might play some role in its GDP-mannose binding?. And also, why hydroxyl group is involved in binding of ScVrg4 but not in CtVrg4 toward GDP-mannose? Both proteins have similar structural fold (line 133, RMSD 1.6A⁰) despite of 53% sequence identity?. The authors should provide more detail for this or may be possible reason for difference in binding properties?

2. Line 196: Based on Blot assays, CtVrg4∆31 mutation also shows binding with similar lipids as the Wild type protein. Comparing blots in Fig 6a, it looks like protein after mutation losing its specificity as showing slight binding with all other lipids too? Wild type proteins bind more specifically toward mentioned lipids. Secondly, the authors compare the lipid binding sites of CtVrg4 with ScVrg4 but did not provide explanation what could be the reason behind the differences?

Minor points

3. Line 55, The line does not go with explanation provided in the paragraph.

6. PLOS authors have the option to publish the peer review history of their article (what does this mean?). If published, this will include your full peer review and any attached files.

Reviewer #1: No

Reviewer #2: No

---

## [Author Response · Author response to Decision Letter 0]

2 Mar 2023

We thank the reviewers for their valuable comments. We have now modified the manuscript as shown below. The questions from the reviewers are in red. The responses are in black. The material in the manuscript is in quotes.

Reviewer #1 -

1. The assays and results are described in a clear manner, although no clear rationale is observed across the different experiments, in order to understand why each assay was performed or how each results connects to another. For example, why not analyze the CtVrg4 cleaved form (cCtVrg4) on the complementation assay with Hygromycin B or the CtVrg4d31 version. In the same way, why do not probe the CtVrg4d17 on the lipid assay in parallel to the CtVrg4d31 deletion mutant.

Response : 

The idea was to show that the transporter we purified was functional in both in vivo and in vitro transport assays. The hygromycin B assay and the proteoliposome assay are hence complementary techniques. Concerning the mutants – we did a complementation assay with Hygromycin B to observe the impact of single amino acid deletions on the protein's functionality. We did a bioinformatic analysis to understand the potential chymotrypsin cleaving sites and made the different constructs of CtVrg4 (1- 368 aa, 34-368 aa, and 34-385 aa). The complementation assay showed the rescue phenotype, and this data was not included in the manuscript to avoid the flow of the manuscript. We have demonstrated the kinetics and lipid binding activity of CtVrg4d31. We felt that showing the activity in the complementation assay was redundant. So, we did not do the experiment. In the predicted structure of CtVrg4, the last 17 amino acids are disordered and may not have a direct role in lipid interaction. So, we did not conduct any lipid assay for CtVrg4d17.

2. . Independent of these comments, the major drawback of this work is the lack of biological replicates in the assays described in the figures 5, 6 and 7. Since these experiments lack of biological replicates, authors does not report statical analysis as expect but it impact on the overall interpretation of results. It would be adequate to include more biological replicates and perform an statical analysis to support these observations as the journal requires.

Response: In the methods section of the Transport Assay and the Thermal Shift assay, we state that we have technical and biological repeats. It is also mentioned in Figures 5 and 6 legends. The calculated IC50 and Kd values represent the mean of two independent biological repeats (each done in technical duplicate or triplicate); errors are indicated as S.D (Standard Deviation) in the right bottom/top of the figures. Figure 7 (a,b, and d) also shows error bars from the data; the raw data is the same as what is shown in Figure 5 and Figure 6. See line 351-354 and 379-381 (Manuscript-v2-with-track changes).

Reviewer #2 - 

1. Line 170: Based on binding studies, the authors shows that hydroxyl group Y310 in CtVrg4 is not critical for binding. It makes sense for binding of GMP-mannose with CtVrg4 WT /mutant Y130F as Kd is in same range but for binding of GDP with CtVRg4 WT is stronger compared to mutant as Kd is much lower. The hydroxyl group might play some role in its GDP-mannose binding?. And also, why hydroxyl group is involved in binding of ScVrg4 but not in CtVrg4 toward GDP-mannose? Both proteins have similar structural fold (line 133, RMSD 1.6A⁰) despite of 53% sequence identity? The authors should provide more detail for this or may be possible reason for difference in binding properties?

Response: Thank you for pointing out that this needs to be clarified. We have added a line immediately after Line 170 “We discuss later our idea that the importance of Y310 may come from its role in positioning Y54, which is critical for binding to the ribose sugar.”

Later in the discussion section, we provide an explanation and a hypothesis for the role of Y310.

“In ScVrg4, Y281 forms a hydrogen bond with the ribose sugar of GDP-mannose and a π-stacking interaction with Y28, which along with S269, further coordinates the ribose sugar. The alanine mutants Y28A, and Y281A of ScVrg4, resulted in the complete abolishment of transport activity [11]. In our complementation assay, we found that Y310F (equivalent to Y281 in ScVrg4) conferred a partial rescue, whereas no complementation was observed in the case of Y310A, Y54A, and Y54F. The Kd for GDP-mannose almost doubled in the Y310F mutant compared with WT, but the Kd for GMP remained the same (Figure 7b). The IC50 for GDP-mannose also doubled compared with the WT, further validating the decreased binding efficiency of GDP-mannose in the Y310F mutation. These results suggest that Y310 also forms a hydrogen bond with the ribose sugar and helps position the Y54 residue to bind the ribose sugar, whose substitution is lethal for transport activity.”

2. Line 196: Based on Blot assays, CtVrg4∆31 mutation also shows binding with similar lipids as the Wild type protein. Comparing blots in Fig 6a, it looks like protein after mutation losing its specificity as showing slight binding with all other lipids too? Wild type proteins bind more specifically toward mentioned lipids. Secondly, the authors compare the lipid binding sites of CtVrg4 with ScVrg4 but did not provide explanation what could be the reason behind the differences?

Response: We believe the CtVrg4∆31 mutation does not bind to all the lipids. In Figure 6(a), if we compare the intensity of binding of the blank (bottom right) with all the other spots, we notice no difference in intensity. It is noise – and according to PLOS-ONE policy, we are providing unmodified figures. Changing the intensity of the dark spots to make the blank white will show no non-specific binding. DMPC (experimental proof, Ref 11) and DPPC (MD simulation, Ref 12) were found, as described in previous studies, to bind to ScVrg4. This paper explored new additional lipids that can bind to the CtVrg4. We hope that given the sequence and Alphafold model similarity, we expect that ScVrg4 will bind to PIPs as well. 

3. Line 55, The line does not go with explanation provided in the paragraph.

Response: Remodified phrase of the sentence (lines 54 - 55) – We hope it is now clear.

“Although GDP-mannose is a naturally occurring NS in humans, no NST that transports it is found. Hence, it provides a unique opportunity to target GDP-mannose transporters for fighting fungal infections in humans where mannose is the most abundant sugar of the fungal cell wall, which directly supports the integrity of the cell”.

---

## [Decision Letter · Decision Letter 1]

5 Apr 2023

Biochemical characterization of a GDP-mannose transporter from Chaetomium thermophilum

PONE-D-23-00914R1

Dear Dr. Subramanian,

We’re pleased to inform you that your manuscript has been judged scientifically suitable for publication and will be formally accepted for publication once it meets all outstanding technical requirements.

Kind regards,

Michael Massiah

Academic Editor

PLOS ONE

Additional Editor Comments (optional):

Reviewers' comments:

Reviewer's Responses to Questions

**Comments to the Author**

1. If the authors have adequately addressed your comments raised in a previous round of review and you feel that this manuscript is now acceptable for publication, you may indicate that here to bypass the “Comments to the Author” section, enter your conflict of interest statement in the “Confidential to Editor” section, and submit your "Accept" recommendation.

Reviewer #2: All comments have been addressed

2. Is the manuscript technically sound, and do the data support the conclusions?

Reviewer #2: Yes

3. Has the statistical analysis been performed appropriately and rigorously? 

Reviewer #2: N/A

4. Have the authors made all data underlying the findings in their manuscript fully available?

Reviewer #2: Yes

5. Is the manuscript presented in an intelligible fashion and written in standard English?

Reviewer #2: Yes

6. Review Comments to the Author

Reviewer #2: (No Response)

7. PLOS authors have the option to publish the peer review history of their article (what does this mean?). If published, this will include your full peer review and any attached files.

Reviewer #2: **Yes: **Anupreet kaur

---

## [Editor Report · Acceptance letter]

11 Apr 2023

PONE-D-23-00914R1 

Biochemical characterization of a GDP-mannose transporter from *Chaetomium thermophilum*

Dear Dr. Ramaswamy:

I'm pleased to inform you that your manuscript has been deemed suitable for publication in PLOS ONE. Congratulations! Your manuscript is now with our production department. 

Kind regards, 

on behalf of

Dr. Michael Massiah 

Academic Editor

PLOS ONE